# Automatic Annotation and Evaluation of Error Types for Grammatical Error Correction

## Abstract

Until now, error type performance for Grammatical Error Correction (GEC) systems could only be measured in terms of recall because system output is not annotated. In this paper, we overcome this problem by using a linguistically-enhanced alignment to automatically extract the edits between parallel original and corrected sentences and then classify them using a new dataset-independent rule-based classifier. As human experts rated the predicted error types as "Good" or "Acceptable" in at least 95% of cases, we applied our approach to the system output produced in the CoNLL-2014 shared task to carry out a detailed analysis of system error type performance for the first time.

## 1 Introduction

The Conference on Natural Language Learning (CoNLL) shared task of 2014 (Ng et al., 2014) required teams to build systems that were capable of correcting all types of grammatical errors in learner text. While the submitted systems were evaluated against text that had been explicitly annotated with error type information, the teams themselves were not required to annotate their output in a similar way. This mismatch ultimately meant that a detailed error type analysis of each system was impossible and that error type performance could only be measured in terms of recall.

The main aim of this paper is to rectify this situation and provide a method by which unannotated error correction data can be automatically annotated with error type information. This is important because some systems may be more effective at correcting certain error types than oth-

ers, yet this information is otherwise concealed in an overall score. Although several new metrics and methodologies for Grammatical Error Correction (GEC) have been proposed since the end of the CoNLL-2014 shared task (Felice and Briscoe, 2015; Bryant and Ng, 2015; Napoles et al., 2015; Grundkiewicz et al., 2015), none of these are currently capable of producing individual error type scores.

Our approach consists of two main steps. First, we automatically extract the edits between parallel original and corrected sentences by means of a linguistically-enhanced alignment algorithm (Felice et al., 2016), and second, we classify them according to a new rule-based framework specifically designed with error type evaluation in mind. This enables us to automatically annotate system hypothesis corrections with the same alignment and error type information as the reference and hence carry out a more detailed evaluation. The tool we use to do this will be released with this paper.

## 2 Edit Extraction

The first stage of automatic annotation is *edit extraction*. Specifically, given an original and corrected sentence pair, we need to determine the start and end boundaries of any edits. This is fundamentally an alignment problem:

| We | took | a | **guide** | tour | **on** | | **center** | **city** | . |
|----|------|---|-----------|------|--------|---|------------|----------|---|
| We | took | a | **guided** | tour | **of** | **the** | **city** | **center** | . |

Table 1: A sample alignment between an original and corrected sentence (Felice et al., 2016).

The first attempt at automatic edit extraction was made by Swanson and Yamangil (2012), who simply used the Levenshtein distance to align original and corrected sentence pairs. As the Levenshtein distance only aligns individual tokens how-

ever, they also merged all adjacent non-matches in an effort to capture multi-token edits. Xue and Hwa (2014) subsequently improved on Swanson and Yamangil's work by training a maximum entropy classifier to predict whether edits should be merged or not.

Most recently, Felice et al. (2016) proposed a new method of edit extraction using a linguistically-enhanced alignment supported by a set of merging rules. In particular, they incorporated various linguistic information, such as part-of-speech and lemma, into the cost function of the Damerau-Levenshtein[1] algorithm to make it more likely that tokens with similar linguistic properties align. This approach ultimately proved most effective at approximating human edits in several datasets (80-85 $F_1$), and so we use it in the present study.

## 3 Automatic Error Typing

Having extracted the edits, the next step is to assign them error types. While Swanson and Yamangil (2012) did this by means of maximum entropy classifiers, one disadvantage of this approach is that such classifiers are biased towards their particular training corpora. In particular, the fact that different datasets are annotated according to different standards means that it is inappropriate to evaluate the predicted error types of an in-domain corpus against the predicted error types of an out-of-domain corpus (c.f. Xue and Hwa (2014)). Instead, a dataset-agnostic error type evaluation is much more desirable.

To solve this problem, we took inspiration from Swanson and Yamangil's (2012) observation that most error types are based on part-of-speech (POS) categories, and wrote a rule to classify an edit based only on its automatic POS tags. We then added another rule to differentiate Missing, Unnecessary and Replacement errors depending on whether tokens were inserted, deleted or substituted. From there, we extended our approach to classify errors that are not well-characterised by POS alone (such as Spelling or Word Order) and ensured that all types are assigned based only on automatically obtained properties of the data.

One of the key strengths of this approach is that by being dependent only on automatic mark-up information, our classifier is entirely dataset in-

dependent and does not require labelled training data. This is in contrast with machine learning approaches which require different classifiers for different datasets and which ultimately may not be entirely compatible with each other. Instead, our approach is analogous to automating the annotation guidelines given to human annotators.

A second significant advantage of our approach is that it is also always possible to determine precisely why an edit was assigned to a particular error category. In contrast, human and machine learning classification decisions are often less transparent and may furthermore be subject to annotator bias. Moreover, by being fully deterministic, our approach bypasses bias effects altogether and should hence be more consistent.

### 3.1 Automatic Markup

The prerequisites for our rule-based classifier are that each token in both the original and corrected sentence is POS tagged, lemmatized, stemmed and dependency parsed. We use spaCy[2] v1.6 for all but the stemming, which is performed by the Lancaster Stemmer in NLTK.[3] Since fine-grained POS tags are often too detailed for the purposes of error evaluation, we also map spaCy's Penn Treebank style tags to the coarser set of Universal Dependency tags.[4] We use the latest Hunspell GB-large dictionary[5] to help classify non-word errors. The marked-up tokens in an edit span are then input to our classifier and an error type is returned.

### 3.2 Error Categories

The complete list of 25 error types in our new framework is shown in Table 2. Note that most of them can be prefixed with 'M:', 'R:' or 'U:', depending on whether they describe a Missing, Replacement, or Unnecessary edit, to enable evaluation at different levels of granularity (See Appendix A for all valid combinations). This means we can choose to evaluate, for example, only *replacement* errors (anything prefixed by 'R:'), only *noun* errors (anything suffixed with 'NOUN') or only *replacement noun* errors ('R:NOUN'). This flexibility allows us to make more detailed observations about different aspects of system perfor-

---

[1] Damerau-Levenshtein is an extension of Levenshtein that also handles transpositions; e.g. AB→BA

[2] https://spacy.io/
[3] http://www.nltk.org/
[4] http://universaldependencies.org/tagset-conversion/en-penn-uposf.html
[5] https://sourceforge.net/projects/wordlist/files/speller/2016.11.20/

| Code | Meaning | Description / Example |
|------|---------|----------------------|
| **ADJ** | Adjective | *big → wide* |
| **ADJ:FORM** | Adjective Form | Comparative or superlative adjective errors. *goodest → best, bigger → biggest, more easy → easier* |
| **ADV** | Adverb | *speedily → quickly* |
| **CONJ** | Conjunction | *and → but* |
| **CONTR** | Contraction | *n't → not* |
| **DET** | Determiner | *the → a* |
| **MORPH** | Morphology | Tokens have the same lemma, but nothing else in common. *quick (adj) → quickly (adv)* |
| **NOUN** | Noun | *person → people* |
| **NOUN:INFL** | Noun Inflection | Count-mass noun errors. *informations → information* |
| **NOUN:NUM** | Noun Number | *cat → cats* |
| **NOUN:POSS** | Noun Possessive | *friends → friend's* |
| **ORTH** | Orthography | Case and/or whitespace errors. *Bestfriend → best friend* |
| **OTHER** | Other | Errors that do not fall into any other category (e.g. paraphrasing). *at his best → well, job → professional* |
| **PART** | Particle | *(look) in → (look) at* |
| **PREP** | Preposition | *of → at* |
| **PRON** | Pronoun | *ours → ourselves* |
| **PUNCT** | Punctuation | *! → .* |
| **SPELL** | Spelling | *genectic → genetic, color → colour* |
| **UNK** | Unknown | The annotator detected an error but was unable to correct it. |
| **VERB** | Verb | *ambulate → walk* |
| **VERB:FORM** | Verb Form | Infinitives (with or without "to"), gerunds (-ing) and participles. *to eat → eating, dancing → danced* |
| **VERB:INFL** | Verb Inflection | Misapplication of tense morphology. *getted → got, fliped → flipped* |
| **VERB:SVA** | Subject-Verb Agreement | *(He) have → (He) has* |
| **VERB:TENSE** | Verb Tense | Includes inflectional and periphrastic tense, modal verbs and passivization. *eats → ate, eats → has eaten, eats → can eat, eats → was eaten* |
| **WO** | Word Order | *only can → can only* |

Table 2: The list of 25 main error categories in our new framework with examples and explanations.

mance.

One caveat concerning error scheme design is that it is always possible to add new categories for increasingly detailed error types; for instance, we currently label [*could → should*] a tense error, when it might otherwise be considered a modal error. The reason we do not call it a modal error, however, is because it would then become less clear how to handle other cases such as [*can → should*] and [*has eaten → should eat*], which might be considered a more complex combination of a modal and tense error. As it is impractical to create new categories and rules to differentiate between such narrow distinctions however, our final framework aims to be a compromise between informativeness and practicality.

### 3.3 Classifier Evaluation

As our new error scheme is based only on automatically obtained properties of the data, there are no gold standard labels against which to evaluate classifier performance. For this reason, we instead carried out a small-scale manual evaluation, where we simply asked 5 GEC researchers to rate the appropriateness of the predicted error categories for 200 randomly chosen edits in context (100 from FCE-test (Yannakoudakis et al., 2011) and 100 from CoNLL-2014) as "Good", "Acceptable" or "Bad". "Good' meant the chosen category was the most appropriate for the given edit, "Acceptable" meant the chosen category was appropriate, but probably not optimum, while "Bad" meant the chosen category was not appropriate for the edit. Raters were warned that edit boundaries had been determined automatically, and hence might be unusual, but that they should focus on the appropriateness of the error category regardless of whether they agreed with the boundary or not.

The result of this evaluation is shown in Table 3. Significantly, all 5 raters individually considered at least 95% of our rule-based error types to be either "Good" or "Acceptable", despite the degree of noise introduced by automatic edit extraction. Furthermore, whenever raters judged an edit as "Bad", this could usually be traced back to a mistake made by the POS tagger; e.g. [*ring*

| Rater | Good | Acceptable | Bad |
|-------|------|-----------|-----|
| 1 | 92.0% | 4.0% | 4.0% |
| 2 | 89.5% | 6.5% | 4.0% |
| 3 | 83.0% | 13.0% | 4.0% |
| 4 | 84.5% | 11.0% | 4.5% |
| 5 | 82.5% | 15.5% | 2.0% |
| **OVERALL** | **86.3%** | **10.0%** | **3.7%** |

Table 3: The percent distribution for how each expert rated the appropriateness of the predicted error types. E.g. Rater 3 considered 83% of all predicted types to be "Good".

$\rightarrow$ *rings*] might be considered a NOUN:NUM or VERB:SVA error depending on whether the tagger considered both sides of the edit nouns or verbs. Inter-annotator agreement was good at 0.724 $\kappa_{free}$ (Randolph, 2005).

In contrast, the best results using a classifier were between 50-70 $F_1$ (Felice et al., 2016). Although these results are incomparable with previous approaches which were evaluated using a different metric and error scheme, we nevertheless believe that the high scores awarded by the raters validate the efficacy of our rule-based approach.

## 4 CoNLL-2014 Shared Task Analysis

To demonstrate the value of our approach, we applied our automatic annotation tool to the data produced in the CoNLL-2014 shared task (Ng et al., 2014). In particular, we used our tool to generate annotated versions of the system output files produced by each participating team.[6] Although our approach can be applied to any dataset, we chose CoNLL-2014 because it constitutes the largest collection of publicly available GEC system output.

One benefit of explicitly annotating the hypothesis files is that it makes evaluation much more straightforward. Specifically, if both the hypothesis and reference files are annotated in the same format, we need only compare the edits in each file to produce an F-score. In particular, for a given sentence, any edit with the same span and correction in both files is a true positive (TP), while the remaining edits in the hypothesis are false positives (FP) and the remaining edits in the reference are false negatives (FN). This is in contrast with all other metrics in GEC, which typically incorporate some sort of edit extraction or alignment component directly into their evalua-

---

[6] http://www.comp.nus.edu.sg/~nlp/conll14st.html

|  | M2 Scorer | | Our Approach | |
|------|------|------|------|------|
| Team | Gold | Auto | Gold | Auto |
| AMU | 35.01 | 35.06 | 32.67 | 32.22 |
| CAMB | 37.33 | 37.32 | 34.92 | 33.99 |
| CUUI | 36.79 | 37.64 | 34.15 | 34.68 |
| IITB | 5.90 | 5.97 | 5.77 | 5.74 |
| IPN | 7.09 | 7.69 | 6.12 | 6.15 |
| NTHU | 29.92 | 29.85 | 26.74 | 25.74 |
| PKU | 25.32 | 25.40 | 23.95 | 23.62 |
| POST | 30.88 | 31.02 | 28.43 | 28.00 |
| RAC | 26.68 | 26.89 | 23.39 | 23.21 |
| SJTU | 15.19 | 15.24 | 15.15 | 14.90 |
| UFC | 7.84 | 7.90 | 7.97 | 7.90 |
| UMC | 25.37 | 25.46 | 23.77 | 23.53 |

Table 4: Overall scores for each team in CoNLL-2014 using gold and auto references with both the M2 scorer and our simpler edit comparison approach. All scores are in terms of $F_{0.5}$.

tion algorithms (Dahlmeier and Ng, 2012; Felice and Briscoe, 2015; Napoles et al., 2015). Our approach, on the other hand, treats edit extraction and evaluation as separate tasks.

### 4.1 Gold Reference vs. Auto Reference

Before evaluating the newly annotated hypothesis files against the reference, we must also address another mismatch: namely that the hypothesis edits were aligned and classified automatically, while the reference edits were aligned and classified manually using a different framework. Since evaluation is now a straightforward comparison between two files however, it is especially important that both files are processed in the same way. For instance, a hypothesis edit [*have eating → has eaten*] will not match the reference edits [*have → has*] and [*eating → eaten*] because the former is one edit while the latter is two edits, even though they equate to the same thing.

We can solve this problem by reprocessing the reference file in the same way as the hypothesis file. This means all the reference edits are subject to the same alignment and classification criteria as the hypothesis edits. While it may seem unorthodox to discard gold reference information in favour of automatic reference information, Table 4 shows that this has no significant impact on the results when using either the M2 scorer, the *de facto* standard of GEC evaluation (Dahlmeier and Ng, 2012), or our own approach.

| Type | AMU P | AMU R | AMU $F_{0.5}$ | CAMB P | CAMB R | CAMB $F_{0.5}$ | CUUI P | CUUI R | CUUI $F_{0.5}$ | IITB P | IITB R | IITB $F_{0.5}$ |
|---|---|---|---|---|---|---|---|---|---|---|---|---|
| Missing | 43.61 | 14.36 | 30.98 | 46.32 | 30.00 | 41.77 | 26.71 | 18.62 | 24.57 | 15.38 | 0.59 | 2.56 |
| Replacement | 37.19 | 26.90 | 34.54 | 37.37 | 28.07 | 35.05 | 45.78 | 22.89 | 38.15 | 29.85 | 1.49 | 6.22 |
| Unnecessary | - | - | - | 25.51 | 27.59 | 25.90 | 34.20 | 33.91 | 34.14 | 46.15 | 1.55 | 6.83 |

| Type | IPN P | IPN R | IPN $F_{0.5}$ | NTHU P | NTHU R | NTHU $F_{0.5}$ | PKU P | PKU R | PKU $F_{0.5}$ | POST P | POST R | POST $F_{0.5}$ |
|---|---|---|---|---|---|---|---|---|---|---|---|---|
| Missing | 2.86 | 0.29 | 1.04 | 35.34 | 11.60 | 25.08 | 33.33 | 4.37 | 14.34 | 31.14 | 13.13 | 24.44 |
| Replacement | 9.87 | 3.86 | 7.53 | 27.61 | 19.15 | 25.37 | 29.62 | 18.32 | 26.37 | 33.16 | 19.32 | 29.00 |
| Unnecessary | 0.00 | 0.00 | 0.00 | 34.57 | 16.17 | 28.16 | 0.00 | 0.00 | 0.00 | 26.32 | 33.12 | 27.44 |

| Type | RAC P | RAC R | RAC $F_{0.5}$ | SJTU P | SJTU R | SJTU $F_{0.5}$ | UFC P | UFC R | UFC $F_{0.5}$ | UMC P | UMC R | UMC $F_{0.5}$ |
|---|---|---|---|---|---|---|---|---|---|---|---|---|
| Missing | 1.49 | 0.27 | 0.78 | 62.50 | 4.42 | 17.24 | - | - | - | 40.08 | 23.57 | 35.16 |
| Replacement | 29.50 | 20.87 | 27.25 | 50.54 | 3.43 | 13.47 | 72.00 | 2.64 | 11.52 | 34.62 | 9.69 | 22.87 |
| Unnecessary | 0.00 | 0.00 | 0.00 | 17.65 | 11.51 | 15.95 | - | - | - | 16.89 | 17.33 | 16.98 |

Table 5: Precision, recall and $F_{0.5}$ for Missing, Unnecessary, and Replacement errors for each team. A dash indicates the team's system did not attempt to correct the given error type (TP+FP = 0).

We validated this hypothesis for each team by means of bootstrap significance testing (Efron and Tibshirani, 1993) and found no statistically significant difference between auto and gold references (1,000 iterations, $p > .05$). This leads us to conclude that our auto annotations are qualitatively as good as human annotations.

Table 4 also shows that M2 scores tend to be higher than our own, which initially led us to believe that our approach was underestimating performance. We subsequently found, however, that the M2 scorer in fact tends to overestimate performance (c.f. Felice and Briscoe (2015) and Napoles et al. (2015)).

In particular, given a choice between matching [*have eating → has eaten*] from Annotator 1 or [*have → has*] and [*eating → eaten*] from Annotator 2, the M2 scorer will always choose Annotator 2 because two true positives (TP) are worth more than one. Similarly, whenever the scorer encounters two false positives (FP) within a certain distance of each other,[7] it merges them and treats them as one false positive; e.g. [*is a cat → are a cats*] is selected over [*is → are*] and [*cat → cats*] even though these edits are best handled separately. Ultimately, it can be said that the M2 scorer exploits its dynamic edit boundary prediction in order to maximise true positives and minimise false positives and hence produces slightly inflated scores.

---

[7]The distance is controlled by the *max_unchanged_words* parameter which is set to 2 by default.

### 4.2 Operation Tier

In our first category experiment, we simply investigated the performance of each system in terms of Unnecessary, Missing or Replacement edits. The results are shown in Table 5.

The most surprising result is that 5 teams failed to correctly resolve any unnecessary token errors at all (AMU, IPN, PKU, RAC, UFC). This is especially surprising given that we would expect unnecessary token errors to be easier to correct than others; a system need only detect and delete without having to propose any alternative. There is also no obvious explanation as to why these teams had difficulty with this error type because each of them employed different combinations of correction strategies including machine translation (MT), language modelling, classifiers and rules.

In contrast, CUUI's classifier approach (Rozovskaya et al., 2014) was the most successful at correcting not only unnecessary token errors, but also replacement token errors, while CAMB's hybrid MT approach (Felice et al., 2014) significantly outperformed all others in terms of missing token errors. It would hence make sense to combine these two approaches, and indeed recent research has shown this improves overall performance (Rozovskaya and Roth, 2016).

### 4.3 General Error Types

Table 6 shows precision, recall and $F_{0.5}$ for each of the error types in our proposed framework for each team in CoNLL-2014. We refer the reader to the shared task paper for more information about

| | | AMU | CAMB | CUUI | IITB | IPN | NTHU | PKU | POST | RAC | SJTU | UFC | UMC |
|---|---|---|---|---|---|---|---|---|---|---|---|---|---|
| **ADJ** | P | 7.14 | 9.09 | - | 0.00 | 0.00 | 0.00 | 50.00 | 0.00 | 11.11 | 0.00 | - | 4.35 |
| | R | 9.38 | 12.82 | - | 0.00 | 0.00 | 0.00 | 6.67 | 0.00 | 3.33 | 0.00 | - | 3.57 |
| | F$_{0.5}$ | 7.50 | 9.65 | - | 0.00 | 0.00 | 0.00 | **21.74** | 0.00 | 7.58 | 0.00 | - | 4.17 |
| **ADJ:FORM** | P | 60.00 | 75.00 | 100.00 | 100.00 | 0.00 | 33.33 | 100.00 | 50.00 | 11.54 | - | - | 80.00 |
| | R | 60.00 | 50.00 | 27.27 | 33.33 | 0.00 | 33.33 | 33.33 | 11.11 | 42.86 | - | - | 57.14 |
| | F$_{0.5}$ | 60.00 | 68.18 | 65.22 | 71.43 | 0.00 | 33.33 | 71.43 | 29.41 | 13.51 | - | - | **74.07** |
| **ADV** | P | 11.76 | 12.66 | 0.00 | 0.00 | 0.00 | 0.00 | 0.00 | - | 0.00 | 4.76 | - | 7.27 |
| | R | 5.88 | 23.26 | 0.00 | 0.00 | 0.00 | 0.00 | 0.00 | - | 0.00 | 3.03 | - | 10.00 |
| | F$_{0.5}$ | 9.80 | **13.93** | 0.00 | 0.00 | 0.00 | 0.00 | 0.00 | - | 0.00 | 4.27 | - | 7.69 |
| **CONJ** | P | 6.25 | 0.00 | - | - | 0.00 | 0.00 | 0.00 | - | 0.00 | - | 0.00 | 0.00 |
| | R | 7.69 | 0.00 | - | - | 0.00 | 0.00 | 0.00 | - | 0.00 | - | 0.00 | 0.00 |
| | F$_{0.5}$ | **6.49** | 0.00 | - | - | 0.00 | 0.00 | 0.00 | - | 0.00 | - | 0.00 | 0.00 |
| **CONTR** | P | 29.17 | 40.00 | 46.15 | - | 0.00 | - | - | 33.33 | 0.00 | 66.67 | - | 28.57 |
| | R | 87.50 | 28.57 | 75.00 | - | 0.00 | - | - | 50.00 | 0.00 | 33.33 | - | 28.57 |
| | F$_{0.5}$ | 33.65 | 37.04 | 50.00 | - | 0.00 | - | - | 35.71 | 0.00 | **55.56** | - | 28.57 |
| **DET** | P | 33.55 | 36.44 | 30.92 | 21.43 | 0.00 | 35.91 | 29.35 | 26.12 | 0.00 | 43.88 | - | 36.36 |
| | R | 14.13 | 43.17 | 52.03 | 0.92 | 0.00 | 28.53 | 49.41 | 49.41 | 0.00 | 12.57 | - | 23.72 |
| | F$_{0.5}$ | 26.32 | **37.61** | 33.65 | 3.93 | 0.00 | 34.14 | 18.99 | 28.84 | 0.00 | 29.29 | - | 32.86 |
| **MORPH** | P | 54.67 | 57.35 | 52.94 | 15.38 | 2.25 | 25.00 | 21.43 | 30.30 | 27.27 | 50.00 | 36.36 | 34.48 |
| | R | 45.56 | 45.35 | 20.00 | 2.70 | 2.78 | 20.25 | 32.14 | 12.82 | 15.19 | 2.74 | 5.06 | 11.63 |
| | F$_{0.5}$ | 52.56 | **54.47** | 39.82 | 7.94 | 2.34 | 23.88 | 22.96 | 23.81 | 23.53 | 11.24 | 16.26 | 24.75 |
| **NOUN** | P | 25.35 | 28.42 | 0.00 | 25.00 | 8.33 | 0.00 | 3.45 | 10.00 | 30.43 | 0.00 | - | 28.57 |
| | R | 15.52 | 22.13 | 0.00 | 2.22 | 4.30 | 0.00 | 0.96 | 1.90 | 6.54 | 0.00 | - | 10.00 |
| | F$_{0.5}$ | 22.50 | **26.89** | 0.00 | 8.20 | 7.02 | 0.00 | 2.27 | 5.41 | 17.59 | 0.00 | - | 20.83 |
| **NOUN:INFL** | P | 55.56 | 60.00 | 50.00 | - | 0.00 | 100.00 | 57.14 | 80.00 | 60.00 | 0.00 | - | - |
| | R | 83.33 | 75.00 | 66.67 | - | 0.00 | 40.00 | 57.14 | 66.67 | 60.00 | 0.00 | - | - |
| | F$_{0.5}$ | 59.52 | 62.50 | 52.63 | - | 0.00 | **76.92** | 57.14 | **76.92** | 60.00 | 0.00 | - | - |
| **NOUN:NUM** | P | 48.24 | 42.59 | 43.57 | 43.75 | 12.84 | 44.05 | 28.92 | 30.52 | 27.72 | 54.29 | - | 44.93 |
| | R | 55.66 | 53.00 | 59.91 | 3.95 | 10.05 | 48.54 | 42.34 | 56.54 | 35.92 | 10.50 | - | 17.32 |
| | F$_{0.5}$ | **49.56** | 44.33 | 46.09 | 14.52 | 12.16 | 44.88 | 30.88 | 33.62 | 29.04 | 29.60 | - | 34.07 |
| **NOUN:POSS** | P | 20.00 | 66.67 | - | - | - | - | 20.00 | 0.00 | 0.00 | 25.00 | - | 50.00 |
| | R | 14.29 | 10.53 | - | - | - | - | 5.26 | 0.00 | 0.00 | 4.55 | - | 5.00 |
| | F$_{0.5}$ | 18.52 | **32.26** | - | - | - | - | 12.82 | 0.00 | 0.00 | 13.16 | - | 17.86 |
| **ORTH** | P | 60.00 | 66.67 | 73.81 | - | 3.45 | 0.00 | 28.57 | 49.32 | 16.67 | - | - | 50.00 |
| | R | 11.11 | 40.00 | 59.62 | - | 4.55 | 0.00 | 6.90 | 64.29 | 49.12 | - | - | 17.24 |
| | F$_{0.5}$ | 31.91 | 58.82 | **70.45** | - | 3.62 | 0.00 | 17.54 | 51.72 | 19.20 | - | - | 36.23 |
| **OTHER** | P | 19.33 | 23.57 | 16.13 | 12.50 | 2.38 | 1.40 | 11.11 | 13.95 | 0.00 | 0.00 | - | 13.54 |
| | R | 6.65 | 9.87 | 1.37 | 0.30 | 0.31 | 0.58 | 0.58 | 1.69 | 0.00 | 0.00 | - | 3.74 |
| | F$_{0.5}$ | 13.99 | **18.44** | 5.12 | 1.39 | 1.02 | 1.09 | 2.40 | 5.70 | 0.00 | 0.00 | - | 8.88 |
| **PART** | P | 71.43 | 26.67 | 0.00 | - | - | 12.90 | - | - | - | 40.00 | - | 18.18 |
| | R | 20.00 | 16.00 | 0.00 | - | - | 16.00 | - | - | - | 9.09 | - | 10.00 |
| | F$_{0.5}$ | **47.17** | 23.53 | 0.00 | - | - | 13.42 | - | - | - | 23.81 | - | 15.63 |
| **PREP** | P | 47.62 | 41.70 | 32.69 | 75.00 | 0.00 | 10.95 | - | 21.74 | 0.00 | 37.50 | - | 20.53 |
| | R | 16.53 | 35.91 | 13.65 | 1.44 | 0.00 | 12.81 | - | 2.18 | 0.00 | 7.18 | - | 13.42 |
| | F$_{0.5}$ | 34.60 | **40.40** | 25.56 | 6.67 | 0.00 | 11.28 | - | 7.79 | 0.00 | 20.33 | - | 18.56 |
| **PRON** | P | 43.75 | 20.00 | 0.00 | 0.00 | 11.11 | 50.00 | 100.00 | 27.27 | 4.76 | 0.00 | - | 22.45 |
| | R | 9.72 | 13.25 | 0.00 | 0.00 | 1.72 | 2.86 | 1.56 | 4.76 | 1.54 | 0.00 | - | 14.10 |
| | F$_{0.5}$ | **25.74** | 18.15 | 0.00 | 0.00 | 5.32 | 11.63 | 7.35 | 14.02 | 3.36 | 0.00 | - | 20.07 |
| **PUNCT** | P | 25.00 | 60.47 | 39.53 | 100.00 | 0.00 | 48.28 | - | 27.27 | 0.00 | 5.00 | - | 43.02 |
| | R | 3.57 | 15.66 | 11.33 | 1.85 | 0.00 | 9.72 | - | 6.34 | 0.00 | 0.97 | - | 23.13 |
| | F$_{0.5}$ | 11.36 | **38.46** | 26.40 | 8.62 | 0.00 | 26.92 | - | 16.42 | 0.00 | 2.73 | - | 36.71 |
| **SPELL** | P | 77.78 | 78.43 | 50.00 | 0.00 | 30.77 | 0.00 | 44.58 | 68.27 | 74.60 | - | - | 100.00 |
| | R | 64.95 | 42.55 | 2.60 | 0.00 | 5.41 | 0.00 | 71.15 | 70.30 | 86.24 | - | - | 1.32 |
| | F$_{0.5}$ | 74.82 | 67.11 | 10.75 | 0.00 | 15.87 | 0.00 | 48.18 | 68.67 | **76.67** | - | - | 6.25 |
| **VERB** | P | 18.84 | 16.09 | - | 0.00 | 7.69 | 0.00 | 14.29 | 0.00 | 0.00 | 0.00 | - | 18.87 |
| | R | 8.07 | 8.86 | - | 0.00 | 0.71 | 0.00 | 0.68 | 0.00 | 0.00 | 0.00 | - | 6.58 |
| | F$_{0.5}$ | **14.87** | 13.83 | - | 0.00 | 2.60 | 0.00 | 2.87 | 0.00 | 0.00 | 0.00 | - | 13.74 |
| **VERB:FORM** | P | 34.85 | 38.24 | 70.59 | 50.00 | 8.77 | 36.84 | 30.77 | 20.00 | 35.42 | 30.77 | 100.00 | 34.04 |
| | R | 23.71 | 26.26 | 26.37 | 1.15 | 5.75 | 36.84 | 35.16 | 3.45 | 34.69 | 4.65 | 1.22 | 18.18 |
| | F$_{0.5}$ | 31.86 | 35.04 | **52.86** | 5.26 | 7.94 | 36.84 | 31.56 | 10.20 | 35.27 | 14.49 | 5.81 | 28.99 |
| **VERB:INFL** | P | 100.00 | 100.00 | - | - | 100.00 | 100.00 | 50.00 | 100.00 | 100.00 | - | 0.00 | - |
| | R | 100.00 | 100.00 | - | - | 50.00 | 50.00 | 50.00 | 50.00 | 100.00 | - | 0.00 | - |
| | F$_{0.5}$ | **100.00** | **100.00** | - | - | 83.33 | 83.33 | 50.00 | 83.33 | **100.00** | - | 0.00 | - |
| **VERB:SVA** | P | 49.06 | 42.68 | 54.71 | 50.00 | 24.53 | 50.58 | 57.14 | 33.33 | 34.83 | 59.09 | 82.86 | 58.33 |
| | R | 27.37 | 31.82 | 70.45 | 1.15 | 13.98 | 66.92 | 17.20 | 16.67 | 31.00 | 14.13 | 28.16 | 14.29 |
| | F$_{0.5}$ | 42.35 | 39.95 | 57.27 | 5.26 | 21.31 | 53.18 | 39.02 | 27.78 | 33.99 | 36.11 | **59.67** | 36.08 |
| **VERB:TENSE** | P | 20.55 | 26.05 | 75.00 | 66.67 | 7.14 | 38.89 | 10.61 | 20.00 | 23.27 | 15.38 | 100.00 | 30.51 |
| | R | 8.82 | 17.92 | 5.33 | 1.27 | 1.24 | 4.22 | 4.35 | 2.34 | 21.26 | 2.52 | 1.26 | 10.98 |
| | F$_{0.5}$ | 16.23 | **23.88** | 20.74 | 5.92 | 3.66 | 14.71 | 8.24 | 7.97 | 22.84 | 7.60 | 5.99 | 22.50 |
| **WO** | P | - | 38.89 | 0.00 | 66.67 | - | - | - | 0.00 | 0.00 | - | - | 41.18 |
| | R | - | 33.33 | 0.00 | 14.29 | - | - | - | 0.00 | 0.00 | - | - | 35.00 |
| | F$_{0.5}$ | - | 37.63 | 0.00 | 38.46 | - | - | - | 0.00 | 0.00 | - | - | **39.77** |

Table 6: Precision, recall and F$_{0.5}$ for each team and error type. A dash indicates the team's system did not attempt to correct the given error type (TP+FP = 0). The highest F-score for each type is highlighted.

each team's system (Ng et al., 2014).

Overall, CAMB was the most successful team in terms of error types, achieving the highest F-score in 10 (out of 24) error categories, followed by AMU (Junczys-Dowmunt and Grundkiewicz, 2014), who scored highest in 6 categories. All but 2 teams (IITB and IPN) achieved the best score in at least 1 category, which suggests that different approaches to GEC complement different error types. Only CAMB attempted to correct at least 1 error from every category.

Regarding individual error categories: PKU's language model approach significantly outperformed all others in handling ADJ errors (21.74 $F_{0.5}$). ADV and CONJ errors proved extremely difficult for all teams, with the best results at 13.93 $F_{0.5}$ (CAMB) and 6.49 $F_{0.5}$ (AMU) respectively. Although several teams built specialist classifiers for DET errors, CAMB's hybrid MT system still slightly outperformed them (37.61 $F_{0.5}$). MT approaches were most effective at correcting NOUN errors (AMU, CAMB, UMC), while fairly high scores for NOUN:NUM errors showed that this category could be successfully handled by MT (AMU, CAMB), classifiers (CUUI) or language model approaches (NTHU). Few teams attempted to correct NOUN:POSS errors, but CAMB's system handled them the best (32.26 $F_{0.5}$). CUUI's classifier for ORTH errors significantly outperformed all other teams at 70.45 $F_{0.5}$. As with DET errors, several teams employed specialist classifiers to tackle PREP errors, but CAMB's hybrid system still worked best overall (40.40 $F_{0.5}$). AMU's MT system was most successful at correcting PRON errors (25.74 $F_{0.5}$), while CAMB was most successful at correcting PUNCT errors (38.46 $F_{0.5}$). Although spell checkers are widespread nowadays, many teams did not seem to employ them; this would have been an easy way to boost overall performance. CUUI's classifier approach to VERB:FORM errors significantly outperformed other approaches (52.86 $F_{0.5}$), suggesting a classifier is well-suited to this category. While UFC's rule-based approach achieved the highest score for VERB:SVA errors (59.67 $F_{0.5}$), it is worth noting that CUUI's classifier approach was not far behind (57.27 $F_{0.5}$). Finally, only 3 teams were successful at handling WO errors (CAMB, IITB and UMC), all of whom achieved similar scores of just under 40 $F_{0.5}$ using MT.

| CAMB | | | |
|---|---|---|---|
| Type | P | R | $F_{0.5}$ |
| M:DET | 43.79 | 51.75 | 45.18 |
| R:DET | 19.46 | 35.37 | 21.39 |
| U:DET | 43.75 | 40.10 | 42.97 |
| DET | 36.44 | 43.17 | 37.61 |

| CUUI | | | |
|---|---|---|---|
| Type | P | R | $F_{0.5}$ |
| M:DET | 23.86 | 44.37 | 26.29 |
| R:DET | 27.03 | 24.69 | 26.53 |
| U:DET | 36.19 | 66.97 | 39.85 |
| DET | 30.92 | 52.03 | 33.65 |

Table 7: Detailed breakdown of Determiner errors for two teams.

## 4.4 Detailed Error Types

In addition to analysing general error types, the modular design of our framework also allows us to evaluate error type performance at an even greater level of detail. For example, Table 7 shows the breakdown of Determiner errors for two teams using different approaches in terms of edit operation. Note that this is a representative example of detailed error type performance as an analysis of all error type combinations for all teams would take up too much space.

While CAMB's hybrid MT approach achieved a higher score than CUUI's classifier approach overall (37.61 $F_{0.5}$ vs. 33.65 $F_{0.5}$), our more detailed evaluation reveals that actually CUUI's approach performed better at Replacement Determiner errors than CAMB (26.53 $F_{0.5}$ vs. 21.39 $F_{0.5}$). This shows that even though one approach might be better than another overall, other approaches may still have complementary strengths. In fact the main weakness of CUUI's classifier seems to be that a high recall for missing and unnecessary determiners is counterbalanced by a low precision, which suggests that minimising false positives in these categories is the most obvious avenue for improvement.

## 4.5 Multi Token Errors

Another benefit of explicitly annotating all hypothesis edits is that edit spans become fixed; this means we can evaluate system performance in terms of edit size. Table 8 hence shows the overall performance for each team at correcting multi-token edits, where a multi-token edit is an edit that affects at least two tokens on *either* the source *or*

| Team | P | R | $F_{0.5}$ |
|------|-----|-----|-----|
| AMU | 17.14 | 5.33 | 11.88 |
| CAMB | 27.22 | 17.06 | 24.32 |
| CUUI | 15.69 | 3.67 | 9.48 |
| IITB | 28.57 | 0.94 | 4.15 |
| IPN | 3.33 | 0.47 | 1.51 |
| NTHU | 0.00 | 0.00 | 0.00 |
| PKU | 25.00 | 1.40 | 5.73 |
| POST | 12.77 | 2.82 | 7.48 |
| RAC | 2.96 | 2.82 | 2.93 |
| SJTU | 10.00 | 0.47 | 1.99 |
| UFC | - | - | - |
| UMC | 19.82 | 9.82 | 16.47 |

Table 8: Each team's performance at correcting multi-token edits; i.e. there are at least 2 tokens on one side of the edit.

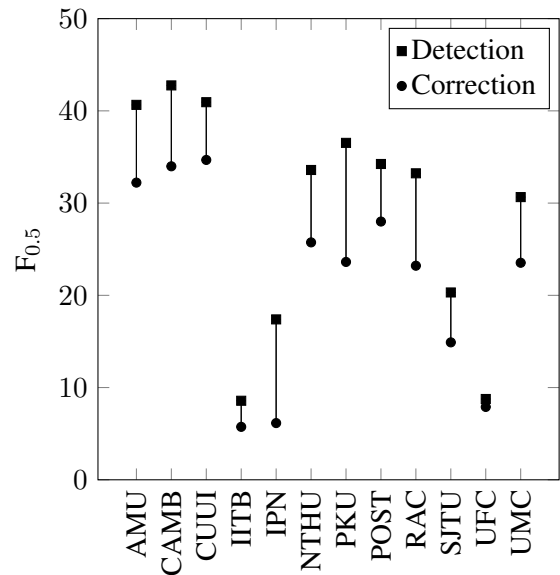

Figure 1: The difference between detection and correction scores for each team overall.

target side. In the CoNLL-2014 test set, there are roughly 220 such edits (about 10% of all edits).

In general, teams did not do well at multi-token edits. In fact only three teams achieved scores greater than 10 $F_{0.5}$ and all of them used MT (AMU, CAMB, UMC). This is significant because recent work has suggested that the main goal of GEC should be to produce fluent-sounding, rather than just grammatical, sentences, even though this often requires complex multi-token edits (Sakaguchi et al., 2016). If no system is particularly adept at correcting multi-token errors however, robust fluency correction will likely require more sophisticated methods than are currently available.

### 4.6 Detection vs. Correction

Another important aspect of GEC that is less frequently reported in the literature is that of error detection; i.e. the extent to which a system can identify erroneous tokens in text. This can be calculated by comparing the edit overlap between the hypothesis and reference files *regardless of the proposed correction* in a manner similar to Recognition evaluation in the HOO shared tasks for GEC (Dale and Kilgarriff, 2011).

Figure 1 hence shows how each team's score for detection differed in relation to their score for correction. While CAMB still scored highest for detection overall, it is interesting to note that the difference between the 2nd and 3rd place contenders (CUUI and AMU) is a lot narrower. This suggests that even though AMU detected roughly as many errors as CUUI, it was less successful at correcting them. IPN and PKU are also notable for detecting

many more errors than they were able to correct.

Although we do not do so here, our scorer is also capable of providing a detailed error type breakdown for detection.

## 5 Conclusion

In this paper, we have described a method to automatically annotate parallel error correction data with explicit edit spans and error type information. This can be used to standardise existing error correction corpora or facilitate a detailed error type evaluation. The tool we use to do this will be released with this paper.

Our approach makes use of previous work to align sentences based on linguistic intuition and then introduces a new rule-based framework to classify edits. This framework is entirely dataset independent, and relies only on automatically obtained information such as POS tags and lemmas. A small-scale evaluation of our classifier found that each rater considered >95% of the predicted error types as either "Good" (85%) or "Acceptable" (10%).

We demonstrated the value of our approach by carrying out a detailed evaluation of system error type performance for the first time for all teams in the CoNLL-2014 shared task on Grammatical Error Correction. We found that different systems had different strengths and weaknesses which we hope researchers can exploit to further improve general performance.

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

# A Complete list of valid error code combinations

| | | Type | Operation Tier | | |
|---|---|---|---|---|---|
| | | | **Missing** | **Unnecessary** | **Replacement** |
| Token Tier | Part Of Speech | Adjective | M:ADJ | U:ADJ | R:ADJ |
| | | Adverb | M:ADV | U:ADV | R:ADV |
| | | Conjunction | M:CONJ | U:CONJ | R:CONJ |
| | | Determiner | M:DET | U:DET | R:DET |
| | | Noun | M:NOUN | U:NOUN | R:NOUN |
| | | Particle | M:PART | U:PART | R:PART |
| | | Preposition | M:PREP | U:PREP | R:PREP |
| | | Pronoun | M:PRON | U:PRON | R:PRON |
| | | Punctuation | M:PUNCT | U:PUNCT | R:PUNCT |
| | | Verb | M:VERB | U:VERB | R:VERB |
| | Other | Contraction | M:CONTR | U:CONTR | R:CONTR |
| | | Morphology | - | - | R:MORPH |
| | | Orthography | - | - | R:ORTH |
| | | Other | M:OTHER | U:OTHER | R:OTHER |
| | | Spelling | - | - | R:SPELL |
| | | Word Order | - | - | R:WO |
| Morphology Tier | | Adjective Form | - | - | R:ADJ:FORM |
| | | Noun Inflection | - | - | R:NOUN:INFL |
| | | Noun Number | - | - | R:NOUN:NUM |
| | | Noun Possessive | M:NOUN:POSS | U:NOUN:POSS | R:NOUN:POSS |
| | | Verb Form | M:VERB:FORM | U:VERB:FORM | R:VERB:FORM |
| | | Verb Inflection | - | - | R:VERB:INFL |
| | | Verb Agreement | - | - | R:VERB:SVA |
| | | Verb Tense | M:VERB:TENSE | U:VERB:TENSE | R:VERB:TENSE |

Table 9: There are 55 total possible error types. This table shows all of them except UNK, which indicates an uncorrected error. A dash indicates an impossible combination.

