# Peer review of "Automatic Annotation and Evaluation of Error Types for Grammatical Error Correction"

_ACL 2017 — decision unknown_

[Official Review · Reviewer 1 · rating 2 · confidence 3]
soundness 4 · originality 3 · clarity 4 · impact 3 · substance 2 · appropriateness 4 · meaningful comparison 3 · presentation format Poster

- Strengths: Useful application for teachers and learners; supports
fine-grained comparison of GEC systems.

- Weaknesses: Highly superficial description of the system; evaluation not
satisfying.

- General Discussion:

The paper presents an approach of automatically enriching the output of GEC
systems with error types. This is a very useful application because both
teachers and learners can benefit from this information (and many GEC systems
only output a corrected version, without making the type of error explicit). It
also allows for finer-grained comparison of GEC systems, in terms of precision
in general, and error type-specific figures for recall and precision.

Unfortunately, the description of the system remains highly superficial. The
core of the system consists of a set of (manually?) created rules but the paper
does not provide any details about these rules. The authors should, e.g., show
some examples of such rules, specify the number of rules, tell us how complex
they are, how they are ordered (could some early rule block the application of
a later rule?), etc. -- Instead of presenting relevant details of the system,
several pages of the paper are devoted to an evaluation of the systems that
participated in CoNLL-2014. Table 6 (which takes one entire page) list results
for all systems, and the text repeats many facts and figures that can be read
off the table. 

The evaluation of the proposed system is not satisfying in several aspects. 
First, the annotators should have independently annotated a gold standard for
the 200 test sentences instead of simply rating the output of the system. Given
a fixed set of tags, it should be possible to produce a gold standard for the
rather small set of test sentences. It is highly probable that the approach
taken in the paper yields considerably better ratings for the annotations than
comparison with a real gold standard (see, e.g., Marcus et al. (1993) for a
comparison of agreement when reviewing pre-annotated data vs. annotating from
scratch). 
Second, it is said that "all 5 raters individually considered at least 95% of
our rule-based error types to be either “Good” or “Acceptable”".
Multiple rates should not be considered individually and their ratings averaged
this way, this is not common practice. If each of the "bad" scores were
assigned to different edits (we don't learn about their distribution from the
paper), 18.5% of the edits were considered "bad" by some annotator -- this
sounds much worse than the average 3.7%, as calculated in the paper.
Third, no information about the test data is provided, e.g. how many error
categories they contain, or which error categories are covered (according to
the cateogories rated as "good" by the annotators).
Forth, what does it mean that "edit boundaries might be unusual"? A more
precise description plus examples are at need here. Could this be problematic
for the application of the system?

The authors state that their system is less domain dependent as compared to
systems that need training data. I'm not sure that this is true. E.g., I
suppose that Hunspell's vocabulary probably doesn't cover all domains in the
same detail, and manually-created rules can be domain-dependent as well -- and
are completely language dependent, a clear drawback as compared to machine
learning approaches. Moreover, the test data used here (FCE-test, CoNLL-2014)
are from one domain only: student essays.

It remains unclear why a new set of error categories was designed. One reason
for the tags is given: to be able to search easily for underspecified
categories (like "NOUN" in general). It seems to me that the tagset presented
in Nicholls (2003) supports such searches as well. Or why not using the
CoNLL-2014 tagset? Then the CoNLL gold standard could have been used for
evaluation.

To sum up, the main motivation of the paper remains somewhat unclear. Is it
about a new system? But the most important details of it are left out. Is it
about a new set of error categories? But hardly any motivation or discussion of
it is provided. Is it about evaluating the CoNLL-2014 systems? But the
presentation of the results remains superficial.

Typos:
- l129 (and others): c.f. -> cf.
- l366 (and others): M2 -> M^2 (= superscribed 2)
- l319: 50-70 F1: what does this mean? 50-70%?

Check references for incorrect case
- e.g. l908: esl -> ESL
- e.g. l878/79: fleiss, kappa

[Official Review · Reviewer 2 · rating 2 · confidence 5]
soundness 4 · originality 3 · clarity 4 · impact 3 · substance 3 · appropriateness 5 · meaningful comparison 3 · presentation format Oral Presentation

The paper presents a novel approach for evaluating grammatical error
correction (GEC) systems. This approach makes it possible to assess
the performance of GEC systems by error type not only in terms of
recall but also in terms of precision, which was previously not
possible in general since system output is usually not annotated with
error categories.

Strengths:

 - The proposed evaluation is an important stepping stone for
   analyzing GEC system behavior.
 - The paper includes evaluation for a variety of systems.
 - The approach has several advantages over previous work:
   - it computes precision by error type
   - it is independent of manual error annotation
   - it can assess the performance on multi token errors
 - The automatically selected error tags for pre-computed error spans
   are mostly approved of by human experts

Weaknesses:

 - A key part – the rules to derive error types – are not described.
 - The classifier evaluation lacks a thorough error analysis and based
   upon that it lacks directions of future work on how to improve the
   classifier.
 - The evaluation was only performed for English and it is unclear how
   difficult it would be to use the approach on another language.

Classifier and Classifier Evaluation
====================================

It is unclear on what basis the error categories were devised. Are
they based on previous work?

Although the approach in general is independent of the alignment
algorithm, the rules are probably not, but the authors don't provide
details on that.  The error categories are a major part of the paper
and the reader should at least get a glimpse of how a rule to assign
an error type looks like.

Unfortunately, the paper does not apply the proposed evaluation on
languages other than English.  It also does not elaborate on what
changes would be necessary to run the classifier on other languages. I
assume that the rules used for determining edit boundaries as well as
for determining the error tags depend on the language/the
pre-processing pipeline to a certain extent and therefore need to be
adapted. Also, the error categories might need to be changed.  The
authors do not provide any detail on the rules for assigning error
categories (how many are there overall/per error type? how complex are
they?) to estimate the effort necessary to use the approach on another
language.

The error spans computed in the pre-processing step seem to be
inherently continuous (which is also the case with the M2 scorer), which
is problematic since there are errors which can only be tagged
accurately when the error span is discontinuous. In German, for
example, verbs with separable prefixes are separated from each other
in the main clause: [1st constituent] [verb] [other constituents]
[verb prefix]. Would the classifier be able to tag discontinuous edit
spans?

The authors write that all human judges rated at least 95\% of the
automatically assigned error tags as appropriate "despite the degree
of noise introduced by automatic edit extraction" (295). I would be
more cautious with this judgment since the raters might also have been
more forgiving when the boundaries were noisy. In addition, they were
not asked to select a tag without knowing the system output but could
in case of noisy boundaries be more biased towards the system
output. Additionally, there was no rating option between "Bad (Not
Appropriate)" and "Appropriate", which might also have led raters to
select "Appropriate" over "Bad". To make the evaluation more sound,
the authors should also evaluate how the human judges rate the
classifier output if the boundaries were manually created,
i.e. without the noise introduced by faulty boundaries.

The classifier evaluation lacks a thorough error analysis. It is only
mentioned that "Bad" is usually traced back to a wrong POS
tag. Questions I'd like to see addressed: When did raters select
"Bad", when "Appropriate"? Does the rating by experts point at
possibilities to improve the classifier?

Gold Reference vs. Auto Reference
=================================

It is unclear on what data the significance test was performed
exactly. Did you test on the F0.5 scores? If so, I don't think this is
a good idea since it is a derived measure with weak discriminative
power (the performance in terms of recall an precision can be totally
different but have the same F0.5 score). Also, at the beginning of
Section 4.1 the authors refer to the mismatch between automatic and
reference in terms of alignment and classification but as far as I can
tell, the comparison between gold and reference is only in terms of
boundaries and not in terms of classification.

Error Type Evaluation
=====================

I do not think it is surprising that 5 teams (~line 473) failed to correct
any unnecessary token error. For at least two of the systems there is
a straightforward explanation why they cannot handle superfluous
words. The most obvious is UFC: Their rule-base approach works on POS
tags (Ng et al., 2014) and it is just not possible to determine
superfluous words based on POS alone. Rozovskaya & Roth (2016) provide
an explanation why AMU performs poorly on superfluous words.

The authors do not analyze or comment the results in Table 6 with
respect to whether the systems were designed to handle the error
type. For some error types, there is a straight-forward mapping
between error type in the gold standard and in the auto reference, for
example for word order error. It remains unclear whether the systems
failed completely on specific error types or were just not designed to
correct them (CUUI for example is reported with precision+recall=0.0,
although it does not target word order errors). In the CUUI case (and
there are probably similar cases), this also points at an error in the
classification which is neither analyzed nor discussed.

Please report also raw values for TP, FP, TN, FN in the appendix for
Table 6. This makes it easier to compare the systems using other
measures. Also, it seems that for some error types and systems the
results in Table 6 are based only on a few instances. This would also
be made clear when reporting the raw values.

Your write "All but 2 teams (IITB and IPN) achieved the best score in
at least 1 category, which suggests that different approaches to GEC
complement different error types." (606) It would be nice to mention
here that this is in line with previous research.

Multi-token error analysis is helpful for future work but the result
needs more interpretation: Some systems are probably inherently unable
to correct such errors but none of the systems were trained on a
parallel corpus of learner data and fluent (in the sense of Sakaguchi
et al, 2016) corrections.

Other
=====

- The authors should have mentioned that for some of the GEC
  approaches, it was not impossible before to provide error
  annotations, e.g. systems with submodules for one error type each.
  Admittedly, the system would need to be adapted to include the
  submodule responsible for a change in the system output. Still, the
  proposed approach enables to compare GEC systems for which producing
  an error tagged output is not straightforward to other systems in a
  unified way.
- References: Some titles lack capitalizations. URL for Sakaguchi et
  al. (2016) needs to be wrapped. Page information is missing for
  Efron and Tibshirani (1993).

Author response
===============

I agree that your approach is not "fatally flawed" and I think this review
actually points out quite some positive aspects. The approach is good, but the
paper is not ready.

The basis for the paper are the rules for classifying errors and the lack of
description is a major factor.        This is not just a matter about additional
examples. If the rules are not seen as a one-off implementation, they need to
be described to be replicable or to adapt them.

Generalization to other languages should not be an afterthought.  It would be
serious limitation if the approach only worked on one language by design.  Even
if you don't perform an adaption for other languages, your approach should be
transparent enough for others to estimate how much work such an adaptation
would be and how well it could reasonably work.  Just stating that most
research is targeted at ESL only reinforces the problem.

You write that the error types certain systems tackle would be "usually obvious
from the tables".  I don't think it is as simple as that -- see the CUUI
example mentioned above as well as the unnecessary token errors.  There are
five systems that don't correct them (Table 5) and it should therefore be
obvious that they did not try to tackle them. However, in the paper you write
that "There
is also no obvious explanation as to why these teams had difficulty with this
error type".